# ELAC2 Functions as a Key Gene in the Early Development of Placental Formation Based on WGCNA

**DOI:** 10.3390/cells12040613

**Published:** 2023-02-14

**Authors:** Junyong Liang, Jingjie Liang, Qiang Tan, Zhengguang Wang

**Affiliations:** 1College of Animal Sciences, Zhejiang University, Hangzhou 310000, China; 2Hainan Institute, Zhejiang University, Sanya 572000, China

**Keywords:** placenta, WGCNA, ELAC2, EMT

## Abstract

The placenta plays a crucial role in mammalian fetal growth. The most important cell type in the placenta is the trophoblast cell. Many genes have been reported to play important functions in the differentiation of early placental trophoblast cells. Weighted gene co-expression network analysis (WGCNA) is a systematic biological method for describing the correlation patterns among genes across microarray samples. We used WGCNA to screen placental trophoblast development-related genes, and through experimental confirmation, we showed that, among these genes, ELAC2 may play an important regulatory role in the early development of mammalian placental formation. ELAC2 regulates early placental trophoblast differentiation by affecting cell migration and cell proliferation. In addition, ELAC2 may be involved in regulating cell migration processes in a manner that affects epithelial mesenchymal transition (EMT).

## 1. Introduction

The process of mammalian fetal growth is complex and regulated by multiple factors. In addition to the growth potential conferred by genes, it is further regulated by maternal effect, the placenta and the environment. The placenta plays a crucial role in maternal–fetal contact [1] and exchange [2]. It is also an important source of pregnancy-related hormones and growth factors [3]. Defects in the placenta have been shown to be the leading cause of reproductive failure [4]. Such defects are associated with many reproductive syndromes [5], such as pre-eclampsia [6], intrauterine growth restriction (IUGR) [7] and chromosomal abnormalities [8].

The most important cell type in the placenta is the trophoblast cell. After the blastocyst stage, the placenta establishes effective contact with the embryo through the villous trophoblast cells. During this period, trophoblast cells further differentiate into cells with different functions. The formation and differentiation of trophoblast cells in the early placenta is considered to be a key event in the formation of a normal placenta, a process that involves substantial cell proliferation and migratory activity [9]. Many genes (e.g., TEAD4 [10], Cdx2 [11], Esrrb [12], Sox2 [13] and Dlx3 [14]) have been reported to function in the differentiation of early placental trophoblast cells. The CDX2 is confirmed to be involved in the lineage development of placental trophoblast cells. The VGLL 1 and its transcription factor TEAD4 were identified as placental trophoblast specific markers. Esrrb functions in the maintenance of placental trophoblast stem cells. The Sox2 is a transcription factor required for the maintenance of pluripotency and plays an essential role in placental trophoblast stem cells. The Dlx3 gene is expressed in the labyrinthine trophoblast of the chorioallantoic placentais and required for the maintenance of Esx1 expression, normal placental morphogenesis, and embryonic survival. Transcriptome analysis has shown that other genes may be involved in the regulation of this process, but their functions remain yet to be investigated.

Weighted gene co-expression network analysis (WGCNA) is a systematic biological method for describing the correlation patterns among genes across microarray samples. WGCNA can be used to find modules of highly correlated genes, summarizing such clusters using the module eigengene or an intramodular hub gene, relating modules to one another and to external sample traits, and calculating module membership measures [15]. Gene modules are detected in the analysis based on the TOM matrix. The significant modules based on the correlation of the modules with the traits are selected for further analysis. Based on the characteristics of the gene distribution in the module, the target genes are verified by constructing the network interoperation map. This bioinformatics analysis method was applied to the genetic screening of multiple samples [16,17,18]. 

In this study, we searched for genes that may play important roles from different periods of placental growth and development using WGCNA, and explored the regulatory mechanisms of the target genes at the cellular level. ELAC2 is identified as a gene involved in placenta formation. Existing studies show that the ELAC2 gene can regulate prostate cells through the TGF-beta/Smad induction pathway [19]. The mechanism of ELAC2 in the field of placental formation remains to be explored.

## 2. Materials and Methods

### 2.1. Identification of Placental Target Genes

The published GSE100053 [20] dataset was used for this experiment based on R (4.1.2). Sequencing data from mice placental samples from different periods in the database were used for data analysis. The WGCNA package was used for data processing. The gene modules were validated for screening by constructing gene co-expression networks. On the basis of the obtained modules, a network graph was constructed to screen the target genes.

### 2.2. Animal Treatments 

Healthy C57BL/6 mice were purchased from the Laboratory Animal Center of Zhejiang University (Hangzhou, China). The mice were housed in an environment with a controlled light cycle (12 h light/12 h darkness), constant temperature and humidity, regular ventilation and free access to food and water. Males (8- to 10-week-old) and females (6- to 8-week-old) were caged in the evening (6:00 p.m.) at a ratio of 1:1 to induce mating. The morning of vaginal plug visualization was designated as day 0.5 of pregnancy (D0.5). The placentas from D8.5, D10.5, D12.5, D14.5 and D16.5 were collected for RNA and protein evaluation. Samples from each group had at least six biological replicates. On the designated day, mice were euthanized through cervical dislocation to collect the placenta. The complete placenta of mice with placental membranes removed was used for the subsequent experiments. All methods were carried out in accordance with the approved protocol and relevant regulations, and complied with the ARRIVE guidelines.

### 2.3. RNA Extraction and RT-qPCR

Total RNA was extracted from the placenta and HTR-8/Svneo cells using TRIzol (Beyotime Biotechnology, Shanghai, China) according to the manufacturer’s instructions. The quantity of RNA was examined using a NanoDrop2000 instrument (Thermo Fisher Scientific, Waltham, MA, USA). For mRNA detection, total RNA (2 μg) was used for cDNA synthesis using a Hieff^®^ UNICON qPCR SYBG Green Master Mix (High Rox) (Yeasen Biotechnology, Shanghai, China). Gene expression was assessed by qPCR with 2 μL of the synthetized cDNA using a Hieff^®^ qPCR SYBR Green Master Mix (High Rox Plus) (Yeasen Biotechnology). The qPCR reactions were performed on a StepOnePlus™ Real-Time PCR System (Applied Biosystems Inc., Foster City, CA, USA). GAPDH was set as the normalizing control. Relative quantities were calculated using the 2−△△CT method. The sequences of all primers used are listed in Table 1.

### 2.4. Cell Culture and Transfection

HTR-8/Svneo cells were purchased from the Cell Bank of the Chinese Academy of Science (Shanghai, China) and cultured in plastic flasks with 5% CO_2_ in air at 37 °C. HTR-8/Svneo cells were seeded in RPMI 1640 medium (Solarbio Science, Beijing, China). All the media used were supplied with 10% FBS (Gibco, Shanghai, China), 100 U penicillin (Sigma Aldrich, Shanghai, China), and 100 μg streptomycin (Solarbio Science). The tranfection of ELAC2 siRNA (GenePharma, Shanghai, China) was performed using Lipofectamine 2000 (Solarbio Science). See the product description for the specific dilution rtio. Cell was transfected in plastic flasks with 5% CO_2_ in air at 37 °C. The cells were collected 48 h after transfection for further study.

### 2.5. Western Blot Analysis

Protein lysates were derived from tissues and cultured cells using RIPA buffer (Beyotime Biotechnology, Shanghai, China) supplemented with 1 mM phenylmethylsulfonyl fluoride (Beyotime Biotechnology). The protein concentrations were detected using an Enhanced BCA Protein Assay Kit (Beyotime Biotechnology). The absorbance values were detected by using a microplate reader. Protein samples need to be denatured before the experiment. The lysates were subjected to 10% sodium dodecyl sulfate–polyacrylamide gel electrophoresis (SDS-PAGE) and transferred to PVDF membranes (Millipore-Sigma, Burlington, MA, USA). The membranes were then blocked in 5% non-fat milk powder in PBS-Tween and incubated with primary antibodies against ELAC2 (1:500, A7128, ABclonal Technology, Wuhan, China), E-cadherin (1:5000, 20874-1-AP, Proteintech Group, Wuhan, China) and vimentin (1:500, CY5134, Abways Technology, Shanghai, China) overnight at 4 °C. After being washed with PBST, the membranes were incubated with horseradish peroxidase (HRP)-conjugated secondary antibodies (1:5000, ABclonal Technology) for 2 h at 37 °C and visualized by chemiluminescent detection using an ECL kit (Beyotime Biotechnology). The original pictures of the Western Blot were analyzed by Image J software (1.53t, NIH, Bethesda, MD, USA).

### 2.6. Wound Healing Assay

Wound healing assays were carried out in 6-well plates. After 48 h transfection, cells were scratched vertically on the cell surface. The cells were washed 3 times with PBS, and a serum-free medium was added to the wells. The scratch closure was investigated under the microscope, and the image of 0 h and 48 h was observed through a microscope. The ratio of the initial scratch area and the final scratch area were analyzed by Image J software (1.53t, NIH, Bethesda, MD, USA).

### 2.7. Transwell Cell Migration Assay

Transwell cell migration assays were carried out in 12-well plates. Transwell chambers (8.0 µm, CORNING, Shanghai, China) were placed in plates. The transfected cells (1 × 105 cells) were suspended in RPMI 1640 without serum and added to the upper chambers. The lower chamber contained a medium supplemented with 10% fetal bovine serum. After 12 h, the cells were fixed with 4% paraformaldehyde (Solarbio Science) and stained with crystal violet staining solution (Beyotime Biotechnology). The resulting images were observed through a microscope. The migrated cells were counted using Image J software (1.53t, NIH, Bethesda, MD, USA).

### 2.8. EdU Cell Proliferation Assay

EdU cell proliferation assays were carried out in 6-well plates. After 48 h transfection, cells were cultured with EdU (Beyotime Biotechnology, Shanghai, China) solution for 2 h and fixed by 4%paraformaldehyde. After removing 4% paraformaldehyde, cells were washed multiple times with washing solution (Beyotime Biotechnology, Shanghai, China). After removal of the washing solution, the cells were immersed with a permeabilized solution (Beyotime Biotechnology, Shanghai, China) and left at room temperature for 15 min. After removing the permeabilization, the cells were washed multiple times with the washing solution. Click reaction solution (Beyotime Biotechnology, Shanghai, China) is prepared according to the product instructions. Hoechst dye (Beyotime Biotechnology Invitrogen) were used to stain nuclei. The EdU-labeled cells were imaged with confocal microscope. The cells were counted using Image J software (1.53t, NIH, Bethesda, MD, USA).

### 2.9. Statistics

All experiments were presented as means ± SDs. Statistical differences between the two groups were analyzed using the two-tailed unpaired Student’s *t*-test. Comparison among multiple groups was conducted using one-way analysis of variance (ANOVA) followed by Dunnett’s test. Statistical significance was defined as *p* < 0.05.

## 3. Results

### 3.1. Construction of the Co-Expression Network

The sample clustering dendrograms was constructed. The soft-power threshold was determined by the function “sft$powerEstimate”; power = 7 was selected for further analysis. Then, gene modules were detected based on the TOM matrix. A total of 17 modules was detected in the analysis. (Figure 1). The most significant yellow modules based on the correlation of the modules with the traits were selected for further analysis (Figure 1). The 794 target genes are included within the yellow module, and the distribution of genes within the yellow module is shown in Figure 2.

### 3.2. Functional Analysis of the Selected Genes

GO analysis was used to identify biological processes enriched in the yellow module. As shown in Figure 3A, the top five biological processes are regulation of hemopoiesis, muscle system process, cellular calcium ion homeostasis, wound healing and the cytokine-mediated signaling pathway. As shown in Figure 3B, the top five molecular functions are receptor ligand activity, guanyl nucleotide binding, guanyl ribonucleotide binding, cytokine activity and phosphoprotein binding. KEGG was used to identify the potential pathways in yellow module. As shown in Figure 3C, the top five pathways are cytokine–cytokine receptor interaction, the chemokine signaling pathway, the JAK-STAT signaling pathway, the Wnt signaling pathway and the toll- receptor signaling pathway.

### 3.3. ELAC2 Was Screened as a Key Gene for Placental Development

Based on the characteristics of the gene distribution in the yellow module, we set the threshold of gene significance in the module to 0.8 and the threshold of the module members to 0.8. Thus, we obtained 25 genes for further study (MAN2A2, LRRC3, PTPN2, SNX33, MRAS, MAP2K3, PRSS22, BC056929, STRAP, IRX1, RXRG, ZBTB26, ELAC2, RNF24, PAK1, RFTN2, FGR, CCL5, PPIL5, ARHGEF19, AMPD3, BCL2, BC055811, DEF6, PRAMEAL13, PPARD, SLC6A13) (Figure 2). By constructing the network interoperation map with 25 target genes (Figure 4), we selected four genes (RFTN2, RNF24, ELAC2 and ZBTB26) located at key nodes for validation. We examined the expression levels of the four genes in the placental tissues of mice at different stages of pregnancy. qPCR results showed significant differences in the expression of ELAC2, which was consistent with the sequencing results (Figure 5).

### 3.4. Cell Migration and Cell Proliferation Are Reduced When ELAC2 Was Downregulated

We down-regulated the expression level of ELAC2 in HTR-8/Svneo cells by transfection with siRNA, and the interference effect was confirmed at both the RNA (Figure 6A) and protein levels (Figure 6B). The results of the wound healing assay showed a significant reduction in the cell healing area after 48 h of interference with ELAC2 compared to the control group (Figure 7A,B). In addition, the results of the transwell cell migration assay showed that the number of cells undergoing migration was significantly reduced after the down-regulation of ELAC2 expression (Figure 7C,D). These results suggest that the inhibition of ELAC2 decreases the migratory ability of the cells. The results of the EdU cell proliferation assays showed that the proportion of EdU-stained HTR-8/Svneo cells was significantly reduced by ELAC2 silencing (Figure 8), suggesting a decrease in the ability of proliferation by ELAC2 silencing.

### 3.5. Interfering with ELAC2 Expression Possibly Promotes Epithelial Mesenchymal Transition

Marker proteins in cell migration were measured in order to elucidate specific mechanisms. The relative protein expression of EMT markers was also measured by Western blot analysis. WB results showed that interference with ELAC2 downregulated the expression of E-cadherin (Figure 6C), an epithelial marker, and induced the expression of vimentin (Figure 6D), a mesenchymal marker, indicating that silencing ELAC2 prompted EMT in trophoblast cells.

## 4. Discussion

In this study, we searched for genes that may play important roles from different periods of placental growth and development using WGCNA. A total of 17 modules was detected in the analysis based on the TOM matrix. The most significant yellow modules based on the correlation of the modules with the traits were selected for further analysis. Based on the characteristics of the gene distribution in the yellow module, we obtained 25 genes for further study. The four target genes (RFTN2, RNF24, ELAC2 and ZBTB26) associated with placental development were verified by constructing the network interoperation map. The ELAC2 was selected for further study by the qPCR validation. There is consistency between the qPCR validation results and sequencing data that ELAC2 expression was significantly higher in the early stage of placenta formation than in the later stage of placenta formation. The main research models for the placental trophoblast level iclude placenta-derived transformed trophoblast cell lines represented by HTR8 SV/ neo and choriocarcinoma-derived cell lines represented by JEG3. The placenta-derived tranformed trophoblast cell lines represented by HTR8SV/neo was selected in this study. HTR8/Svneo [21,22], a human chorionic villous cell line that can mimic early placental trophoblast formation and development, was chosen as the experimental model. HTR-8/Svneo is derived from the trophoblast layer, ease of manipulation, and gain-and-loss-of-function studies are possible. Given the trend of high expression of ELAC2 in early placental formation and low expression in middle and late stages, we used siRNA interference technology to reduce the expression of ELAC2 and then investigated its effect on cell function.

Some enrichments were found by the GO functional analysis and the KEGG signaling pathway analysis. The Ca^2+^ signals have both global and local regulatory effects on cell motility. Guidance receptor signaling is crucial for steering migrating cells [23]. The JAK/STAT [24] pathway and Wnt [25] pathway functions in cell proliferation. At the same time, cell migration and cell proliferation are extensively involved in the formation and differentiation of placental trophoblast cells [9]. Therefore, we chose to perform cell migration and cell proliferation experiments. After the successful reduction in ELAC2 expression by siRNA technology [26], siRNA-treated HTR-8/Svneo resulted in a significant decrease in cell proliferation ability. The results of both the wound healing assay and the transwell cell migration assay confirmed a significant decrease in the migration ability of siRNA-treated HTR-8/Svneo. Extensive proliferative activity exists in placental trophoblast cells for maintaining cellular activity and self-renewal capacity, and to provide support for cell differentiation. By regulating the ability of cell proliferation, ELAC2 may be effectively involved in maintaining the function of placental trophoblast cells. Placental trophoblast cells are involved in maternal–embryonic communication [27], during which they undergo cell migration biological processes such as angiogenesis and chorionic villogenesis [28]. The trophoblast cells eventually differentiate into various subtypes of trophoblast cells distributed to the placental surface to perform specific functions that require the involvement of cell migration. ELAC2 may regulate signaling pathways involved in the cell proliferation migration of the placental trophoblast cells.

Marker proteins in cell migration were measured in order to elucidate specific mechanisms. Epithelial mesenchymal transition (EMT) is a biological process that allows a polarized epithelial cell to acquire greater migratory capacity and invasiveness through various interactions [29]. Vimentin [30] and E-cadherin [31] are markers of EMT. The relative expression of vimentin protein increased in siRNA-treated HTR-8/Svneo. The vimentin protein is a protein involved in the maintenance of the cytoskeleton and its increase enhances the stability of the cytoskeleton [32]. Both EMT and cell migration require active remodeling of the cytoskeleton [33]. The relative expression of the E-cadherin protein decreased in siRNA-treated HTR-8/Svneo. E-cadherin protein is a major component of epithelial cell adhesion junctions and maintains cell adhesion and epithelial cell polarity [34]. Its decrease reduces the ability of EMT and cell migration. The changes in vimentin protein and E-cadherin protein simultaneously confirmed that EMT was affected. The candidate genes affect cell adhesion or polarity that are essential for EMT has to be verified, but the existing results suggest that EMT is highly likely to be involved. We therefore suggest that ELAC2 may be involved in regulating cell migration in a manner that affects EMT.

ELAC2 is widely expressed in multiple tissues [35] and existing studies point to ELAC2 affects biological processes in multiple pathways, mainly including transfer RNA processing [36], interaction with microtubulin [37] and the modification of the transforming growth factor-β (TGF-β) pathway [19]. ELAC2 was shown to have tRNA 3’ endonuclease activity in the C-terminal region and thus to be involved in protein synthesis [38]. In addition, it has been shown to be involved in editing functional small RNAs [39]. This may provide an insight into the role of ELAC2 in regulating the function of placental trophoblast. TGF-β is involved in regulating cell differentiation and wound healing [40]. Our results confirm that changes in ELAC2 expression affect the migratory ability of cells, and whether this phenomenon is affected by TGF-β activity requires further confirmation. However, the study of the specific molecular mechanism of ELAC2 is limited by various conditions and needs to be developed accordingly. Previous studies on ELAC 2 focused on prostate cells and prostate cancer, and this study revealed the effects of ELAC 2 on placenta formation and placental trophoblast cell formation. Research on the mechanism of ELAC 2 in placental trophoblast cells can be further enhanced and improved. Such a complex topic requires a well-explained study to make the material usable for future studies. Although early placental formation and development are very similar in mice and humans [41], and ELAC2 expression is conserved, it would certainly be better to resort to a more uniform and stable model, such as an artificial placenta [42]. 

## 5. Conclusions

In this study, we used WGCNA to screen placental trophoblast development-related genes, combined with experiments to confirm that, among them, ELAC2 may play an important regulatory role in the early development of mammalian placental formation. ELAC2 may regulate early placental trophoblast differentiation by affecting cell migration and proliferation. In addition, ELAC2 may be involved in regulating cell migration processes in a manner that affects EMT. Our findings provide a theoretical basis for further insights into the regulating mechanism of placental development and function.

## Figures and Tables

**Figure 1 cells-12-00613-f001:**
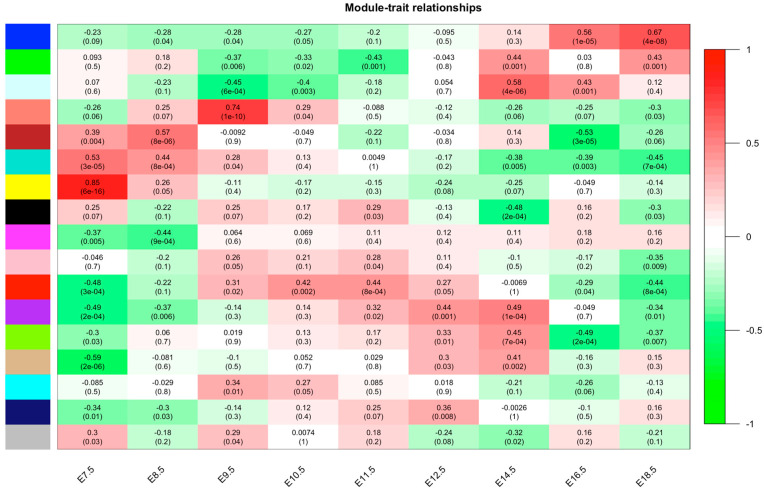
Module–trait relationships. Each row presents a module eigengene, each column presents a clinical trait. Each cell contains the corresponding correlation and *p* value. The table is color-coded by correlation according to the color legend.

**Figure 2 cells-12-00613-f002:**
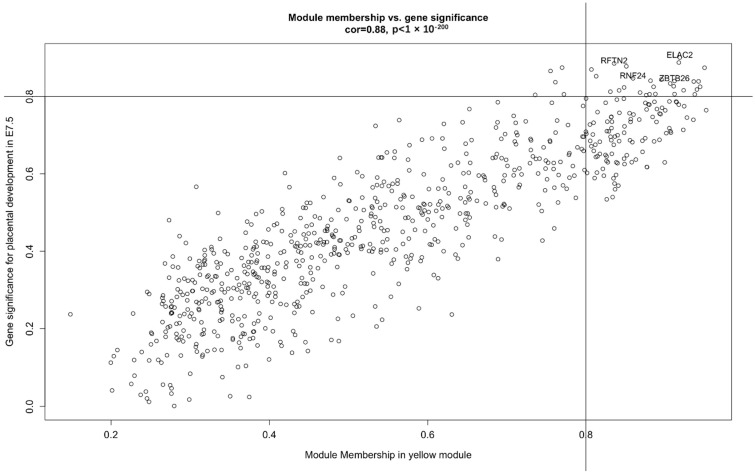
A scatterplot of gene significance (GS) for placental development in E7.5 vs. module membership (MM) in yellow module. The correlation coefficient and *p* value are listed above the scatterplots.

**Figure 3 cells-12-00613-f003:**
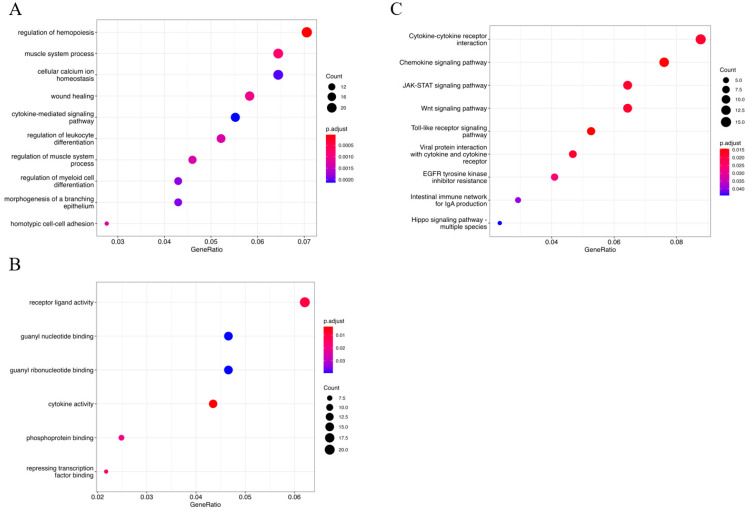
Function enrichment of the genes in the yellow module. The size of the nodes denotes the number of selected genes in pathway or biological processes and the color of the nodes refers to the significance of the results. (**A**) GO-BP (biological processes); (**B**) GO-MF (molecular function); (**C**) KEGG pathway.

**Figure 4 cells-12-00613-f004:**
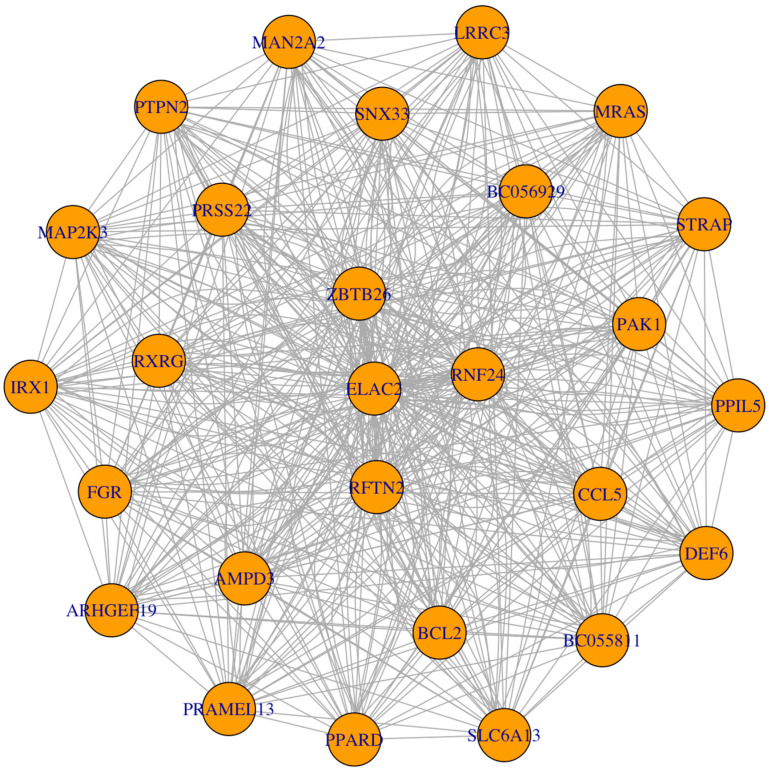
Visualization of the co-expression of genes in the co-expression module. The dot is the gene node, and the line connecting the two dots indicates the interaction between two genes.

**Figure 5 cells-12-00613-f005:**
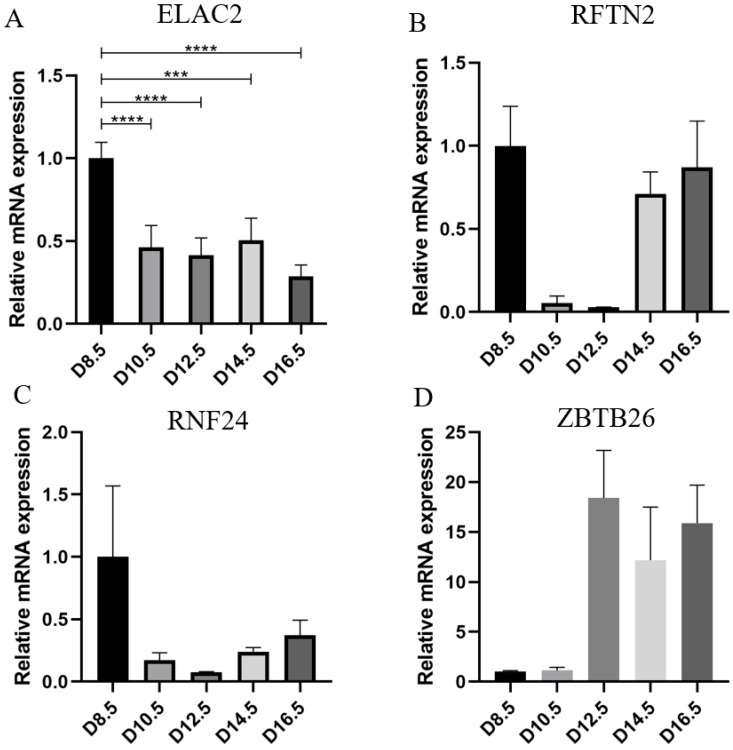
Detection of gene expression in mouse placenta at different developmental stage. (**A**) Detection of ELAC2 mRNA expression by qPCR. (**B**) Detection of RFTN2 mRNA expression by qPCR. (**C**) Detection of RNF24 mRNA expression by qPCR. (**D**) Detection of ZBTB26 mRNA expression by qPCR. *** *p* < 0.001, **** *p* < 0.0001.

**Figure 6 cells-12-00613-f006:**
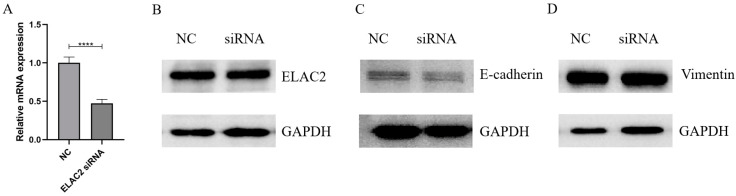
Interference with ELAC2 downregulated EMT-related genes. (**A**) Detection of ELAC2 mRNA expression by qPCR. (**B**) Detection of ELAC2 protein expression by WB. (**C**) Detection of E-cadherin protein expression by WB. (**D**) Detection of vimentin protein expression by WB. **** *p* < 0. 0001. The bolts presented here were chopped; original blots can be found in Appendix A.

**Figure 7 cells-12-00613-f007:**
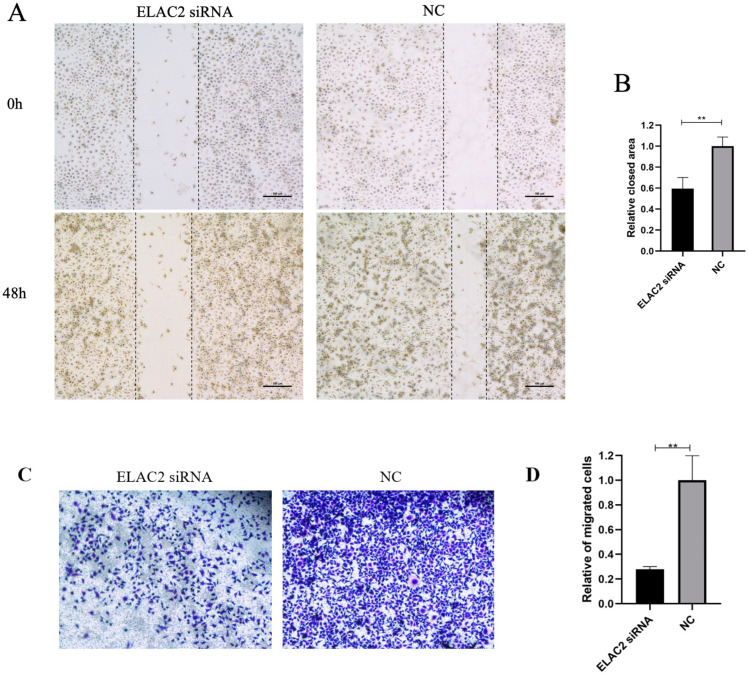
Downregulation of ELAC2 interfere with cell migration in HTR-8/Svneo. (**A**) Representative image of wound healing assay. (**B**) Results of software calculation of relative closed area in cells. (**C**) Representative image of transwell migration assay. (**D**) Results of software calculation of relative migration cells. ** *p* < 0.01.

**Figure 8 cells-12-00613-f008:**
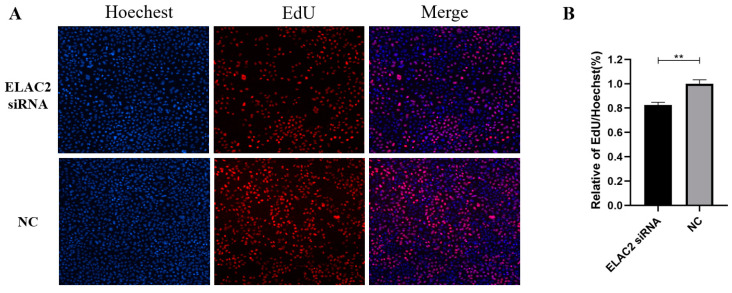
Downregulation of ELAC2 interfere with cell proliferation in HTR-8/Svneo. (**A**) Representative image of EdU staining assay. (**B**) Comparison of EdU signals in si-ELAC2 and control group. ** *p* < 0.01.

**Table 1 cells-12-00613-t001:** Primer sequences for RT-qPCR.

Gene Symbol	Primer Sequence (5′-3′)
GAPDH	F: AGGTCGGTGTGAACGGATTTG
	R: TGTAGACCATGTAGTTGAGGTCA
RFTN2	F: ACCTAAAGGGGACCAGTTACC
	R: ACAATGGCGTCGTTTCCTCTC
RNF24	F: TCCAGAATCTGCCTCTCAACA
	R: CCAGTTCATCTCTAGGCTTGAAG
ELAC2	F: GAGAAGGCGTCCAACGACTTA
	R: AGAAAGATGTTGTCCAAGCGAG
ZBTB26	F: TTGACTGCTGCGAGTTTTCTT
	R: CTGCTGTTCTTTCGACTGGGG

## Data Availability

Not applicable.

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
