# Peer review of "ELAC2 Functions as a Key Gene in the Early Development of Placental Formation Based on WGCNA"

_cells, 2023, doi:10.3390/cells12040613_

Round 1

Reviewer 1 Report

Liang et al have report here in this manuscript that ELAC2 gene as identified as one of the  key genes from the WGCNA , may play a critical role in the migration of trophoblasts during the early development of placental formation. The manuscript overall presents novel findings, however the research methodology, results and discussion lacks clarity. My comments are listed as follows:

1. In the introduction it was unclear whether the authors were focusing on human or murine placental development. The first paragraph indicated the study could be focused on human placental development, however in line 31- of the second paragraph the authors were describing placental labyrinth, which is indicative of murine placental development. Could the authors clearly indicate species specific structural details of the placenta.

2. It was also unclear in the second para trophoblast differentiation is referred to specifically for migration and proliferation. Detailed information on the different phenotypes of the differentiated trophoblasts and their role in maternal fetal interface would assist the readers.

3. In the methodology section - it was unclear which placental areas from the murine placenta used for the gene and protein expression analysis.

4. There was no indication of use of the human trophoblast derived cell line in the introduction, and the passage number details are missing in the methodology section. What is the rationale for using HTR8 SV/neo as opposed to JEG3 or other trophoblast derived cell lines.

5. In line 133 - description on the network analysis - is it a typo in the software details?

6. The authors provide successful knockdown for gene expression by qPCR and WB, at the functional level whether the candidate genes affect cell adhesion or polarity that are essential for EMT has to be verified. In the current study, there is no direct evidence to show that the candidate genes involvement in EMT.

7. Throughout the manuscript language editing is critical, there are quite a few places present and past tenses are mixed up. 

Author Response

1. In the introduction it was unclear whether the authors were focusing on human or murine placental development. The first paragraph indicated the study could be focused on human placental development, however in line 31- of the second paragraph the authors were describing placental labyrinth, which is indicative of murine placental development. Could the authors clearly indicate species specific structural details of the placenta.

Thank you very much for your question. This study was launched with the help of mouse placenta samples and human cell lines, so the introduction involves the interpretation of human placenta and mouse placenta, respectively. Mouse and human placentas remained consistent according to common placental differentiation criteria. According to the distribution of villi on the membrane, humans and mice are discal placenta, which are concentrated in the discal area of the chorionic; according to the barrier structure of the placenta, human and mice are blood chorionic placenta, and there is no endometrial epithelium or connective tissue. The fetal chorionic epithelium has direct contact with the blood between the maternal endometrial villi. According to whether the uterine tissue is damaged during delivery, humans and mice are decidual placenta, that is, the placenta penetrates into the maternal endometrium to form the decidua. The decidua falls off with the fetal membranes and damages the uterine tissue. In conclusion, both mouse and human placentas are introduced in part of this manuscript.

2. It was also unclear in the second para trophoblast differentiation is referred to specifically for migration and proliferation. Detailed information on the different phenotypes of the differentiated trophoblasts and their role in maternal fetal interface would assist the readers.

Thank you very much for your question. Placental trophoblast cell differentiation is a systematic and complex process, but some studies have pointed out that there are extensive cell proliferation and cell migration in the process of placental trophoblast cell differentiation, so this study tries to conduct a preliminary exploration from the perspective of cell proliferation and cell migration.

3. In the methodology section - it was unclear which placental areas from the murine placenta used for the gene and protein expression analysis.

Thank you very much for your question. Complete placenta of mice with placental membranes removed was used for subsequent experiments.

4. There was no indication of use of the human trophoblast derived cell line in the introduction, and the passage number details are missing in the methodology section. What is the rationale for using HTR8 SV/neo as opposed to JEG3 or other trophoblast derived cell lines.

Thank you very much for your question. The cell lines used were commercial cells purchased from Cell Bank of Chinese Academy of Science and purchased directly for this study. As you mentioned, the main research models for the placental trophoblast level include Placenta-derived transformed trophoblast cell lines represented by HTR8 SV / neo and Choriocarcinoma-derived cell lines represented by JEG3. Both can be effectively used for the study of placental trophoblast cells, but they have their own advantages and disadvantages. The Placenta-derived transformed trophoblast cell lines are derived from the trophoblast layer, Ease of manipulation, Gain-and loss-of-function studies possible; While Choriocarcinoma-derived cell lines is tumour-derived, this may have greater interference with the gene object-based present study. Based on the advantages and disadvantages of the laboratory, Placenta-derived transformed trophoblast cell lines represented by HTR8 SV / neo was selected in this study.

5. In line 133 - description on the network analysis - is it a typo in the software details?

Thank you very much for your question. This is a display error that was modified.

6. The authors provide successful knockdown for gene expression by qPCR and WB, at the functional level whether the candidate genes affect cell adhesion or polarity that are essential for EMT has to be verified. In the current study, there is no direct evidence to show that the candidate genes involvement in EMT.

Thank you very much for your question. As you have mentioned, cell adhesion experiments or cell polarity are a direct way to demonstrate the involvement of stromal transformation in epithelial cells. But the main aim of this study was to identify an interesting target study gene, so the design in terms of specific mechanisms stayed at a preliminary level. In this study, only the identification of EMT marker proteins was conducted, and it was speculated that EMT may be involved in the changes of two typical EMT marker proteins. Therefore, in the paper, "EMT may be part of the mechanism involved in the regulation of placental trophoblast cell differentiation", and the relevant part needs to be developed later.

7. Throughout the manuscript language editing is critical, there are quite a few places present and past tenses are mixed up.

Thanks very much for your advice. We revisited this section for further refinement.

Reviewer 2 Report

Since an inadequate trophoblast invasion is related to reproductive failure and pregnancy pathologies, it’s interesting to investigate genes that may play essential roles in placental growth, as the authors declared. This is an interesting topic: as the authors note, the way that. And I enjoyed what has the potential to be a very rich data set.

Comments:

-The study design is not clarified in materials and methods. It is undoubtedly preclinical in vivo study but should be declared. Is this a proof-of-concept study?

-Reading the article is difficult due to references in the sentence and the interposition of figures and tables.

Place the references at the end of each paragraph, not in the middle.

-Introduction

ELAC2 is the main gene studied, as indicated by the study's title, but it is not even mentioned in the introduction; what is its role according to the literature?

-Line 36 “Many genes (e.g., CDX211, TEAD412, ESRRB13, SOX214, etc.) have been reported to function in the differentiation of early placental trophoblast cells. Transcriptome analysis has shown that other genes may be involved in the regulation of this process, but their functions remain to be investigated.”

The references are missing. Please add them.

-Results

The findings of the study should be reported with the numerical values (means ± SDs), separated from tables and figures, well written and explained with words, no graphics.

-3.2. Figures, Tables, and Schemes

Figures and tables should be at the end, before references.

Add a quantitative table with means ± SDs. The bar graphs are visually helpful but difficult to interpret.

-Line 221 “HTR-8/Svneo, a human chorionic villous cell line that can mimic early placental trophoblast formation and development, was chosen as the experiment model.”

Please add references. Why was this cell line chosen?

-Line 219: “At the same time, cell migration and cell proliferation are extensively involved in the formation and differentiation of placental trophoblast cells.

Please add references.

Such a complex topic requires a well-explained study to make the material usable for future studies.

Author Response

-The study design is not clarified in materials and methods. It is undoubtedly preclinical in vivo study but should be declared. Is this a proof-of-concept study?

Thank you very much for your question. This study is a proof-of-concept study, mainly aiming to determine a gene involved in the process of placenta formation, and the specific mechanism remains to be developed.

-Reading the article is difficult due to references in the sentence and the interposition of figures and tables.

Place the references at the end of each paragraph, not in the middle.

Thank you very much for your reminder. Modified as recommended

-Introduction

ELAC2 is the main gene studied, as indicated by the study's title, but it is not even mentioned in the introduction; what is its role according to the literature?

Thank you very much for your question. As you mentioned, our introduction to ELAC2 is mainly discussed in the discussion and has been added to the preface according to your suggestions. It should be pointed out that there are few studies on ELAC2 gene, existing studies show that ELAC2 gene can regulate prostate cells through TGF-beta / Smad induction pathway, in addition, ELAC2 may be involved in transcription and transport processes. It follows that this study perhaps identified a direction for the ELAC2 study.

-Line 36 “Many genes (e.g., CDX211, TEAD412, ESRRB13, SOX214, etc.) have been reported to function in the differentiation of early placental trophoblast cells. Transcriptome analysis has shown that other genes may be involved in the regulation of this process, but their functions remain to be investigated.”

The references are missing. Please add them.

Thank you very much for your reminder. Relevant partial references have been added.

-Results

The findings of the study should be reported with the numerical values (means ± SDs), separated from tables and figures, well written and explained with words, no graphics.

 Thank you very much for your reminder. Modified as recommended

-3.2. Figures, Tables, and Schemes

Figures and tables should be at the end, before references.

Add a quantitative table with means ± SDs. The bar graphs are visually helpful but difficult to interpret.

Thank you very much for your reminder. Modified as recommended

-Line 221 “HTR-8/Svneo, a human chorionic villous cell line that can mimic early placental trophoblast formation and development, was chosen as the experiment model.”

Please add references. Why were this cell line chosen?

Thank you very much for your reminder. Relevant partial references have been added.

Thank you very much for your question. The cell lines used were commercial cells purchased from Cell Bank of Chinese Academy of Science and purchased directly for this study. As you mentioned, the main research models for the placental trophoblast level include Placenta-derived transformed trophoblast cell lines represented by HTR8 SV / neo and Choriocarcinoma-derived cell lines represented by JEG 3. Both can be effectively used for the study of placental trophoblast cells, but they have their own advantages and disadvantages. The Placenta-derived transformed trophoblast cell lines are derived from the trophoblast layer, Ease of manipulation, Gain-and loss-of-function studies possible; While Choriocarcinoma-derived cell lines is tumour-derived, this may have greater interference with the gene object-based present study. Based on the advantages and disadvantages of the laboratory, Placenta-derived transformed trophoblast cell lines represented by HTR8 SV / neo was selected in this study.

-Line 219: “At the same time, cell migration and cell proliferation are extensively involved in the formation and differentiation of placental trophoblast cells.

Please add references.

Thank you very much for your reminder. Relevant partial references have been added.

Such a complex topic requires a well-explained study to make the material usable for future studies.

Thank you very much for your reminder. More detailed studies will be conducted in the future.

Round 2

Reviewer 1 Report

The revised manuscript reads well. No further comments to add.

Reviewer 2 Report

The authors have addressed the comments I made in my previous review. I acknowledge the effort made, especially in order to clarify the report of the result section, including tables and figures. Therefore, I have no further comments.